# Analysis of Dibenzyltoluene Mixtures: From Fast Analysis to In-Depth Characterization of the Compounds

**DOI:** 10.3390/molecules28093751

**Published:** 2023-04-27

**Authors:** Xiaolong Ji, Essyllt Louarn, Fabienne Fache, Laurent Vanoye, Anne Bonhommé, Isabelle Pitault, Valérie Meille

**Affiliations:** 1Univ Lyon, Université Claude Bernard Lyon 1, CNRS, IRCELYON, F-69626 Villeurbanne, France; xiaolong.ji@ircelyon.univ-lyon1.fr (X.J.);; 2Univ Lyon, Université Claude Bernard Lyon 1, CNRS, LAGEPP, F-69100 Villeurbanne, France; isabelle.pitault@univ-lyon1.fr; 3Université Paris Saclay, CNRS, ICP, F-91405 Orsay, France; 4Univ Lyon, Université Claude Bernard Lyon 1, CNRS, ICBMS, F-69100 Villeurbanne, France; 5Laboratory of Catalysis, Polymerization, Processes & Materials, Institut de Chimie de Lyon, Université de Lyon, CP2M UMR 5128 CNRS-UCB Lyon 1-CPE Lyon, CPE Lyon 43 Bd du 11 Novembre 1918, F-69616 Villeurbanne, France

**Keywords:** LOHC, DBT, analysis, GC–MS, isomers, synthesis, Raman

## Abstract

The so-called dibenzyltoluene (H0-DBT) heat transfer oil contains numerous isomers of dibenzyltoluene as well as (benzyl)benzyltoluene (methyl group on the central vs. the side aromatic ring). As it is used as a liquid organic hydrogen carrier (LOHC), a detailed analysis of its composition is crucial in assessing the kinetic rate of hydrogenation for each constituent and studying the mechanism of H0-DBT hydrogenation. To identify all of the compounds in the oil, an in-depth analysis of the GC–MS spectra was performed. To confirm peak attribution, we synthesized some DBTs and characterized the pure compounds using NMR and Raman spectroscopies. Moreover, a fast-GC analysis was developed to rapidly determine the degree of hydrogenation of the mixture.

## 1. Introduction

Hydrogen technologies hold significant potential for the transition to carbon-neutral renewable energy systems. However, hydrogen storage and transport present significant challenges. Liquid organic hydrogen carriers (LOHCs) offer a promising solution for long-term, long-distance hydrogen storage that is potentially cheap, safe, and applicable in ambient conditions [1,2]. Using aromatic hydrocarbons or N-containing aromatic molecules, such as H_2_ acceptors, hydrogen storage is achieved through the heterogeneous catalytic hydrogenation of LOHC compounds, with the release of hydrogen typically reached via the dehydrogenation of its H_2_-rich form. This alternative system for hydrogen storage has advantages over conventional compression or liquefaction technologies in terms of storage density, hydrogen release purity, and H_2_ losses over time [3]. Dibenzyltoluene compounds have been studied for hydrogen storage through hydrogenation for over a decade, mainly by the University of Erlangen–Nüremberg (FAU) [4,5]. It is now being studied by many other teams around the world [6,7,8,9]. Many investigations have been published on both the hydrogenation reaction of the H_2_-lean oil (H0-DBT) [10] and the dehydrogenation of the perhydrogenated oil (H18-DBT), covering topics such as thermochemical data [11], catalysts [12,13], and reactors [7,14,15].

However, “dibenzyltoluene” (H0-DBT) is not a pure compound but a mixture of different dibenzyltoluene isomers and some impurities [16]. This mixture is synthesized and commercialized by two industrial suppliers: Eastman (under the trade name Marlotherm SH, formerly produced by SASOL) [16,17] and Arkema (under the trade name Jarytherm) [18]. DBTs are formed via Friedel–Crafts benzylation of toluene with benzyl chloride, leading to preferential para/ortho additions over meta additions [19,20]. The produced mixtures are separated according the number of rings in the molecules, so that most of the two-ring members, benzyltoluene compounds, are separated from the three-ring member DBTs.

However, further purification of the isomers is neither feasible nor desirable for commercial use as a heat transfer fluid due to the DBT isomers’ quite similar physicochemical properties. The DBT mixture contains all six possible isomers of dibenzyltoluene (DBT) and potentially other products such as (benzyl)benzyltoluene (BBT). While all molecules with n${}_{1}$ + n${}_{2}$ = 1 are considered “dibenzyltoluene” according to the generic formula in Figure 1 [18], we will refer to molecules with n${}_{1}$ = 1 and n${}_{2}$ = 0 DBTs, and those with n${}_{1}$ = 0 and n${}_{2}$ = 1 BBTs.

In 2015, a GC/MS analysis of an H0-DBT Marlotherm mix was published, identifying at least 10 GC peaks in the chromatogram [16]. A later study of the Sasol mix identified 8 different peaks on a GCxGC-TOF analysis, with at least 12 peaks present on a 1D-chromatogram. H0-DBT isomers were identified with the molecular ion *m*/*z* 272 in typical mass spectra, with the base peak being *m*/*z* 181 for the major peak identified as 2,4-DBT in the spectra presented by Markiewisz et al. [16], and *m*/*z* 179 in the work done by Modisha et al. [17] linked to the 2,6-DBT structure. However, not all peaks were identified, and except for 2,4-DBT, no attempts were made. In a similar way to what was performed by Kim et al. in the case of benzyltoluene compounds, we considered that synthesizing some molecules could help identify some of the mixture components unambiguously [21].

Some articles also focus on analytical tools used to assess the progress of hydrogenation reactions. Appropriate tools are required to discriminate between all isomers and their hydrogenated counterparts because the reactivities of the different isomers vary to a large extent [8].

After hydrogenation of the H0-DBT mixture at different times, different degrees of hydrogenation can be obtained, leading to a multiplication of the number of molecules (Figure 1). It was demonstrated that the external rings are hydrogenated before the central one using a Ru catalyst [10], but this is not the case for some bimetallic PtM (M = Fe, Ni, Cu) [22]. Various analytical methods have been proposed to estimate the degree of hydrogenation (DoH). Reversed-phase HPLC was used in 2016 to separate the hydrogenated fractions from H0-DBT [23]. The stationary phase was phenyl-hexyl silica and the mobile phase was acetone/water (96/4, *v*/*v*). H0, H6, H12, and H18-DBT were separated in 45 min but all of the isomers of the same DoH presented the same retention times. Moreover, 1H NMR was used to determine the DoH [10]. Although chemical shift predictions for various H0-DBT isomers have been made, no tentative identification of the exact composition of the mixtures was described. Moreover, chemical shift predictions made in the literature did not take into account the presence of BBT species in the mixtures and only discussed DBT molecules.

This paper has two objectives: (i) We developed a fast analytical method to use as a routine analysis to evaluate the DoH, and (ii) performed an in-depth characterization of all of the constituents of the commercial Jarytherm H0-DBT mixture.

## 2. Results and Discussion

### 2.1. Evaluation of DOH Using Fast GC

The analytical method was developed to separate the initial tri-aromatic compounds from the intermediates (bi- and mono-aromatics, cycloalkenes) and the reaction products (cycloalkenes). The most suitable GC phase for this application is the polyethylene glycol (wax) phase; the commercially available column of this type (allowing to work at the highest temperature) is the HeavyWAX column from Agilent [24]. Moreover, as the objective of this study was to conduct a kinetic study that requires high-frequency monitoring of the reaction, we chose a column with a 100-micron internal diameter (fast GC), which reduces the analysis time (three to six times) at the expense of a slight reduction in the analysis’s sensitivity. This is not a problem for hydrogen storage in LOHC since the mixtures are concentrated [25]. The GC chromatogram of the pre-made H0, H6, H12, and H18-DBT mixture is shown in Figure 2.

The retention time attribution of each Hx-DBT family with different degrees of hydrogenation (DoH) was achieved using GC–MS with the same column and method as the GC-FID. By using n-octadecane (C18) as a standard, the quantity of each DBT family ($n_{Hx-DBT}$) could be determined by Equation (Equation 1), where the relative response factors (RRFs) of the Hx-DBT species were supposed to be similar to those of H0-DBT, which was measured using the method presented in the Appendix A; $A_{Hx-DBT}$ represents the area of the Hx-DBT species and $A_{C18}$ represents the area of the standard in a GC chromatogram; $m_{C18}$ represents the mass of C18 and $M_{Hx-DBT}$ represents the molar mass of each DBT family.
(1)$$n_{Hx-DBT}=\frac{RRF_{Hx-DBT}\frac{A_{Hx-DBT}}{A_{C18}}m_{C18}}{M_{Hx-DBT}}$$

Once the quantity of each Hx-DBT family is known, the DoH of a reaction mixture can be determined using Equation (Equation 2).
(2)$$DoH\left(\%\right)=\frac{n_{H6-DBT}+n_{H12-DBT}\times 2+n_{H18-DBT}\times 3}{3\times \sum n_{Hx-DBT}}\times 100\%$$

The fast GC-FID method allowed for a much easier follow-up of the hydrogenation reaction. The evolution of each DBT family and the total DoH of the mixture could be obtained at any given sample time as shown in Figure 3.

### 2.2. Isomer Identification of the H0-DBT Mixture

The H0-DBT mixture is possibly composed of the 15 molecules represented in Figure 4, which are obtained by the Friedel–Crafts (FC) reaction of toluene with benzyl chloride [19,20]. Those containing a central toluene ring belong to the dibenzyltoluene family, while those having an external toluene ring are named x,y${}^{\prime }$-benzyl(benzyltoluene) (x,y${}^{\prime }$-BBT), where x denotes the position of the substituent on the toluene ring relative to the methyl group and y denotes the position on the other ring relative to the xylyl group.

#### 2.2.1. GC–MS Chromatogram

The GC chromatogram of the H0-DBT mixture is presented in Figure 5, and the data are reported in Table 1. The extracted signal of mass over charge *m*/*z* 272 is represented along with the total ion count (TIC) to highlight the presence of the molecular ion of the H0-DBTs. In total, 17 peaks are identified, with peak number (2) being under the detection limit (LOD) of the TIC, peak numbers (1) and (9) being under the quantification limit (LOQ) of the TIC, and peak numbers (3), (5), and (15) being close to the LOQ. To obtain a more concentrated sample, it was injected again, and its chromatogram is presented in the Appendix A, which clearly shows the existence and form of the above-mentioned peaks. On the contrary, seven peaks appear well above the others: (11), (6), (14), (17), (13), (8), and (16), starting from the most concentrated. The main isomer corresponds to peak number (11) with a retention time of 5.370 min.

**Figure 4 molecules-28-03751-f004:**
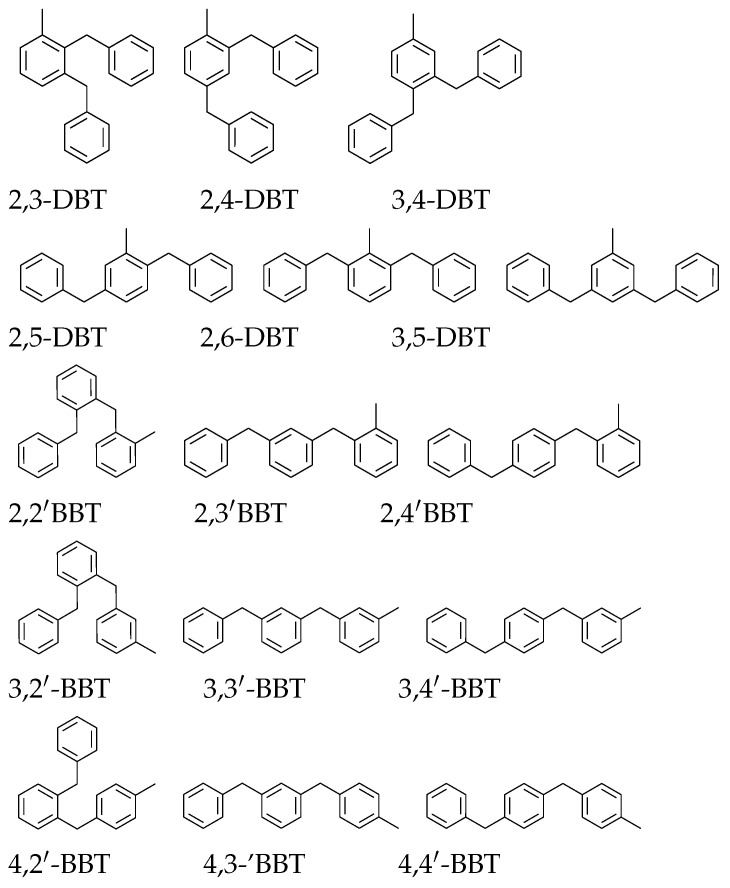
List of DBTs and BBTs present in commercial “DBT” oil.

#### 2.2.2. Friedel–Crafts Reactivity Hypothesis

From the preferential benzylation of the FC reaction, it can be hypothesized that the first benzylation is more likely to occur in the para position compared to the methyl group of the toluene, followed by the ortho position, while the meta position is not favored in this case [18]. Thus, 2,y-DBT and x,4-DBT should predominate. The second benzylation will be selected by the opposing steric effect of the first benzyl position and the reactivity of the FC position, increasing the probability of the meta position compared to the ortho position with respect to the methyl group. The major isomer should be 2,4-DBT followed by 3,4-DBT, 2,6-DBT, and 2,5-DBT. As 3,5-DBT is the fully meta species, its concentration is likely to be the lowest. For the BBTs, if the para position is preferred, the order should be 4,y${}^{\prime }$-BBT > 2,y${}^{\prime }$-BBT, and even > 3,y${}^{\prime }$-BBT; in parallel, x,4${}^{\prime }$-BBT > x,2${}^{\prime }$-BBT > x,3${}^{\prime }$-BBT.

#### 2.2.3. Fractionation Results

Fractionation using an apolar solvent allowed for the separation of two fractions: peaks (1) to (8) ($t_{r}\leq 5.23\;\begin{array}{c} \min \end{array}$) and the later peaks from (9) to (17). It was observed that the first eluted molecules were the same in both apolar (fractionation experiments) and polar (GC–MS column) solvents.

Apolar solvents tend to separate according to the ebullition temperature. The dibenzyl-ortho-substituted species (395–399 °C, as predicted by Advanced Chemistry Development, Inc., ACD/Labs, Software V11.02) generally have a lower boiling point than the other substituted species (399–404 °C, as predicted by ACD/Labs) [20].

The first fraction ($t_{r}<5.23\;\begin{array}{c} \min \end{array}$) should then be due to ortho-substituted compounds such as x,2${}^{\prime }$-BBT, 2,3-DBT and 3,4-DBT.

#### 2.2.4. EI Mass Spectra

Table 1 lists the most important daughter ions obtained for each of the identified retention times. A more detailed table, including the attribution proposed by other authors, is provided in the Appendix A. All species have strong *m*/*z* 272 peaks, corresponding to the molecular ion C_21_H_20_^+·^, indicating stable molecules. However, alkylbenzenes are known for extensive fragmentation, making isomer identification challenging [26]. To the best of our knowledge, there are very few studies describing the spectra of DBT [27,28], and none for the full set of isomers. Katritzky et al. (hereafter K90) attempted to assign GC–MS spectra to DBT-like compounds in a complex mixture of a biomass-based solution [28]. From the mass spectra, they deduced the presence of 4 DBT species (2,4, 3,4, 2,5, and 2,6 DBT) and 5 BBT species (2,2${}^{\prime }$, 4,2${}^{\prime }$, 4,3${}^{\prime }$, 2,4${}^{\prime }$, 4,4${}^{\prime }$-BBT). Meta-substituted compounds were not considered due to the chemical reactivity of the process. Due to their similar spectra, DBTs were difficult to assign in their study. Miranda et al. (hereafter M00) proposed limited information on the spectra of 2,4 and 3,5-DBT and 2,2${}^{\prime }$, 4,2${}^{\prime }$, 2,4${}^{\prime }$ and 4,4${}^{\prime }$-BBT [27]. The two studies did not agree on the species assignment.

**Table 1 molecules-28-03751-t001:** Mass spectra (relative signal for the main observed *m*/*z*) for each peak identified in the chromatogram. The attribution (Attrib.) is related to the retention time ($t_{r}$ in min), peak number(pn), and signal height of the total ion count (TIC) in the chromatogram. The bold represents the attribution confirmed by syntheses.

pn	Attrib.	$t_{r}$	TIC	Ion Mass over Charge *m*/*z*												
		(min)		272	257	195	194	181	180	179	167	166	165	105	104	91
4	2,3 DBT	5.063	82,042	41	2	4	16	31	17	100	4	21	28			17
5	3,2${}^{\prime }$ BBT	5.083	30,024	60			12	47	83	100	23	25	58			27
6	**3,4 DBT**	5.115	259,092	51		4	21	50	15	100	5	27	33			17
7	2,2${}^{\prime }$ BBT	5.188	70,612	12				19	100	85	7	13	36	4	6	14
8	4,2${}^{\prime }$ BBT	5.220	121,124	31			4	21	84	100	9	15	38	6		11
9	**3,5 DBT**	5.293	<LOQ	62				100								
10	2,3${}^{\prime }$ BBT	5.343	40,338	94	10			14	21	23	80	26	56	33	100	46
11	**2,4 DBT**	5.370	621,877	51	4	1	4	100	3	15	4	29	30			31
12	3,4${}^{\prime }$ BBT	5.438	52,165	73	7		5	100	8	23	44	35	45	19		47
13	**2,5 DBT**	5.608	138,433	56	4		4	100	5	30	5	32	33			42
14	2,6 DBT	5.695	228,073	50	1	1	2	100	3	14	5	32	31			31
15	4,3${}^{\prime }$ DBT	5.750	33,111	74				98			100	56	51			35
16	2,4${}^{\prime }$ BBT	5.803	118,206	100	15			98	8	20	89	43	65	24	95	54
17	4,4${}^{\prime }$ BBT	5.843	157,482	75	13		2	100	6	14	61	33	46	16	5	34

From our dataset in Table 1, three main groups can be distinguished from the base and main peaks: (i) the first group (peak numbers 1 to 3) consists of rapidly eluted species with very low concentrations, below LOQ for two of them, whose base peak is *m*/*z* 257 (loss of CH_3_); (ii) a second group of five compounds (peak numbers 4 to 8) with a base peak at *m*/*z* 179 or *m*/*z* 180; (iii) a larger group of compounds is eluted last with a base peak at *m*/*z* 181, except for peak number 10, whose base peak is *m*/*z* 104. Groups (ii) and (iii) represent the two fractions collected, where group (ii) is due to the benzyl-ortho-substituted species, and group (iii) is due to the other species. In the M00 dataset, two groups were also identified, corresponding to our groups (ii) and (iii).

Spectra of groups (ii) and (iii) have the presence of ions *m*/*z* 166 and *m*/*z* 165 in common. Ion *m*/*z* 166 represents the loss of a xylene molecule, which is observed in all molecules. The spectra of dibenzylbenzene (C_20_H_18_) [29] also exhibit similar patterns, which could be attributed to the loss of the central ring after extensive recombination. Ion *m*/*z* 167 (C_6_H_5_−CH^+^−C_6_H_5_ ion), on the other hand, is significantly higher for peak numbers (5), (10), (12), (15), (16), and (17) compared to the other species. This ion is due to the loss of a xylyl radical, a process favored in BBT compared to DBT, as the extra methyl group is located on the side rings. Furthermore, according to Kück’s work on alkylbenzenes [26], *m*/*z* 165 is due to the loss of H_2_ from the *m*/*z* 167 ion by cyclization between the two rings. In fact, the *m*/*z* 167 ion is more important at a low EI energy (below 30 eV) than *m*/*z* 165, as suggested by the author. Therefore, the spectra with a low *m*/*z* 167 ion can be attributed to DBT isomers.

The most interesting features are typically caused by the even-electron daughter ions, as they result from structure-specific rearrangement reactions. In addition to *m*/*z* 166, three even-electron species are identified in the spectra: *m*/*z* 194 (loss of benzene), *m*/*z* 180 (loss of toluene), and *m*/*z* 104 (loss of a double-ringed molecule). These even-electron ions are all due to the McLafferty rearrangement (McL) (see Figure 2 as an example). The structure of the molecule is of great importance in explaining the prevalence of this reaction scheme. In DBTs and BBTs, the McL reaction is initiated by a 1,5-hydrogen transfer from either the methyl or methylene group, which preferably occurs in the ipso site of the ionized aromatic ring. This specificity leads to the increased probability of the McL rearrangement for ortho-substituted species. If the substitution is a phenyl group, it gives access to ion *m*/*z* 180 (x,2${}^{\prime }$-BBT) or ion *m*/*z* 194 (3,4-DBT, 2,3-DBT, and less predominantly, x,2${}^{\prime }$-BBT). If the substitution is a methyl group (2,x${}^{\prime }$-BBT), the ion *m*/*z* 104 is formed (peak numbers (10) and (16)) [27,28]. The specific formation of *m*/*z* 104 has been described for ortho-methyldiphenylmethane [26,30]. It should be noted that *m*/*z* 180 is largely favored over the other McL rearrangements when more than one path is possible, as observed, for example, in 2,2${}^{\prime }$-BBT.

#### 2.2.5. Synthesis and Final Attribution

From the fractionation experiment, it is possible to separate the ortho-benzyl substituted species (4) to (8) from the other substitutions (9) to (17). From the FC reaction, it can be hypothesized that the major peak is 2,4-DBT. This is confirmed by the synthesis of 2,4-DBT, whose mass spectra and retention time are identical to peak number (11). Peak numbers (13) and (14) are assumed to be DBT from the absence of *m*/*z* 167 compared to the other peaks in the same group. The synthesis of 2,5-DBT allowed for the identification of peak number (13), and then by inference, peak number (14) is 2,6-DBT. As expected from the FC reaction, 3,5-DBT identified by synthesis was present at a very low concentration compared to the other DBTs. The last two DBTs were identified in group (ii) by the presence of *m*/*z* 194 and the absence of *m*/*z* 180, which were peak numbers (4) and (6). Again, from the FC reactivity, the para-substituted species should be more concentrated than the ortho-substituted species. The synthesis of 3,4-DBT led to identifying (6) as 3,4-DBT and (4) as 2,3-DBT.

For the BBTs, equivalent arguments for the FC response led to the hypothesis that the prevalence of the substituted species was in the order of 4,y${}^{\prime }$ > 2,y${}^{\prime }$ > > 3,y${}^{\prime }$-BBT. Then peaks (8), (7), and (5) were 4,2${}^{\prime }$-BBT, 2,2${}^{\prime }$-BBT, and 3,2${}^{\prime }$-BBT, respectively. The presence of a strong peak of *m*/*z* 104 led to the conclusion that (10) was 2,3${}^{\prime }$-BBT and (16) was 2,4${}^{\prime }$-BBT. The last eluted compound (17) was likely 4,4${}^{\prime }$-BBT.

No 3,3${}^{\prime }$-BBT was detected, as its formation is less likely and should be well below the detection limit.

## 3. Materials and Methods

### 3.1. Experimental Instrumentation

All solvents and chemicals were of reagent grade and used without any purification. The H0-DBT used in the experiment was a commercial product from Jarytherm (Arkema). Petroleum ether (PE) refers to hydrocarbons (C6–C8) with a boiling range of 40–60 °C. Silica gel (0.040–0.063 nm) was used for column chromatography. NMR analyses were performed at 293 K using either a 300 MHz spectrometer (Bruker AMX 300) or a 500 MHz spectrometer (Bruker DRX 500). HRMS experiments were performed on an Exactive GC Orbitrap (Thermo Fisher, Waltham, MA, USA) using a TriPlus RSH autosampler. The chromatogram and spectra were analyzed using LabSolutions software. The GC-FID analysis was conducted using a HeavyWAX column (10 m × 0.1 mm × 0.1 µm), with helium as the carrier gas at a flow rate of 1 mL/min. The temperature program was as follows: the initial temperature was 200 °C, followed by a ramp from 20 °C·min${}^{-1}$ to 290 °C, which was held for 2 min. The total method time was 6.5 min. Fast GC–MS analyses were performed on a Shimadzu QP2010SE equipped with an Agilent HeavyWAX column (10 m × 0.1 mm × 0.1 μm) in the split mode (carrier gas: H_2_-inj. vol. 0.2 µL-split ratio: 500). The GC temperature program was set as follows: 190 °C (0.5 min)-15 °C/min up to 220 °C (0.5 min)-25 °C/min up to 290 °C (0.3 min). The Raman spectra were recorded at ambient temperatures on a LabRAM HR Evolution spectrometer (HORIBA) equipped with a CCD detector. A diode-pumped solid-state laser with a wavelength of 532 nm was used for excitation and the spectra were calibrated by means of the Raman peak of Si at 520.7 cm${}^{-1}$. The laser focused on the samples under the microscope so that the size of the analyzed spot was about 2 µm. The power of the incident beam on the sample was 1 mW. The time of acquisition was adjusted according to the intensity of the Raman scattering. The wavenumber values obtained from the spectra are considered accurate, within 4 cm${}^{-1}$. The spectra presented are normalized to the band at 1000 cm${}^{-1}$ after a baseline correction.

### 3.2. Fractionation Conditions

Preliminary tests were carried out via thin layer chromatography (TLC) using petroleum ether (C6–C8) as the eluent, which showed three separated spots under 254 nm UV light, of which, the middle one showed fluorescence under 365 nm UV light. The H0-DBT mixture (1 g) was separated through a conventional flash chromatography column packed with silica gel with a height of 18 cm and a diameter of 3.5 cm, giving a dead volume of 150 mL, using 1.5 L petroleum ether as the eluent under pressure and at ambient temperature. Two fractions with different compositions were obtained.

### 3.3. Synthesis of H6, H12, and H18-Dbt by H0-Dbt Hydrogenation

The hydrogenated DBT species were obtained by catalytic hydrogenation of H0-DBT using a commercial 5 wt-% Pt/C catalyst without further treatment in a 300 mL stainless steel autoclave (Parr Instrument) under 30 bars of hydrogen and a 1000 rpm stirring speed at 250 °C, 250 °C, and 150 °C for H6, H12, and H18-DBT, respectively, with different reaction times. Note that the synthesis of H18-DBT requires a long reaction time (5 h) that is not compatible with the use of high temperatures, which would have generated by-products. The H18-DBT obtained had a purity of over 99.9%, and the H12-DBT obtained had a purity of 90%, but the products rich in H6-DBT contained a mixture of Hx-DBT families.

### 3.4. Synthesis of 3,4- and 3,5-Di(benzyl)toluene

3,4 and 3,5-di(benzyl)toluene were synthesized following Figure 3 [31].

The corresponding di(bromomethyl)toluene (1.39 g, 5 mmol) was dissolved in THF (7 mL), followed by the addition of CuI (66 mg, 0.35 mmol), and the solution was cooled to 0 °C. To this solution, the Grignard reagent (phenylmagnesium bromide, 1 M in THF, 13.3 mL, 13.3 mmol) was added drop-wise under nitrogen, and the temperature was gradually raised to ambient. After the mixture was stirred overnight, aq NH_4_Cl (2 mL) was added and the solvent was evaporated under reduced pressure. Then, the residue was extracted with methylene chloride and the organic phase was washed with water. The extract was dried over MgSO_4_, and the solvent was removed under reduced pressure. Purification by flash chromatography (Petroleum ether/AcOEt-98/2) led to the desired product (oil).

3,4-Di(benzyl)toluene

^1^H NMR (300 MHz, CDCl_3_): δ ppm 7.29–6.96 (m, 13H, arom), 3.92 (s, 4H, CH_2_-Ph), 2.31 (s, 3H, CH3); ^13^C NMR (75 MHz, CDCl_3_): 141.0, 140.8, 138.9, 136.2, 136.1, 131.5, 130.7, 128.9, 128.5, 127.4, 126.1, 126.0, 39.1, 38.8, 21.2 (RMN spectra are in the Appendix A). Isolated yield: 70 % (0.97 g). HRMS was calculated for the molecular ion C_21_H_20_: 272.1565; measured: 272.1560. The Raman spectrum is presented in Figure 6 and the main vibrations are reported in Table 2.

3,5-Di(benzyl)toluene

^1^H NMR (300 MHz, CDCl_3_): δ ppm 7.36–7.23 (m, 10H, arom), 6.94–6.91 (m, 3H, arom), 3.98 (s, 4H, CH_2_-Ph), 2.33 (s, 3H, CH_3_); ^13^C NMR (75 MHz, CDCl_3_): 141.4, 141.2, 138.3, 129.0, 128.5, 127.7, 126.9, 126.1, 41.9, 21.4 (RMN spectra are in the Appendix A). Isolated yield: 86 % (1.2 g). HRMS calculated for the molecular ion C_21_H_20_: 272.1565; measured: 272.1564. The Raman spectrum is presented in Figure 6 and the main vibrations are reported in Table 2.

### 3.5. Synthesis of 2,4- and 2,5-Di(benzyl)toluene

The synthesis was performed according to [32] (Figure 4). To a suspension of the corresponding dibromotoluene (250 mg, 1 mmol) in dry Et_2_O, (4.5 mL) was added slowly at −78 °C, n-BuLi (1.1 M in THF, 1 mL). The solution was warmed slowly to room temperature and stirred over a period of 6 h. Then, at −78 °C, benzaldehyde (100 μL, 1 mmol) was added to the yellow solution and the reaction mixture was allowed to warm to room temperature overnight. The reaction was quenched with brine, the aqueous layer was extracted with Et_2_O and the combined organic phases were dried over MgSO_4_, filtrated, and concentrated. The resultant residue was dissolved in dry methylene chloride (4 mL). Triethylsilane (1.1 mL, 4 mmol) was added and the solution was cooled to 0 °C. BF_3_ · Et_2_O (0.2 mL, 1.5 mmol) was added slowly and the reaction mixture was allowed to warm to room temperature overnight. At 0 °C, the reaction was quenched with aq NaHCO_3_, the organic phase was separated and the aqueous phase was extracted with petroleum ether. The combined organic phases were dried over MgSO_4_, filtered, and concentrated. The whole reaction sequence was repeated; after flash chromatography (Petroleum ether/AcOEt-98/2), the desired product was obtained.

2,4-Di(benzyl)toluene (oil)

^1^H NMR (300 MHz, CD_2_Cl_2_): δ ppm 7.34–7.00 (m, 13H, arom), 4.00 (s, 2H, CH_2_-Ph), 3.98 (s, 2H, CH_2_-Ph), 2.24 (s, 3H, CH_3_); ^13^C NMR (75 MHz, CDCl_3_): 141.6, 140.6, 139.0, 138.8, 134.5, 130.9, 130.6, 128.8, 128.5, 127.1, 126.1, 126.0, 41.7, 39.7, 19.4 (RMN spectra are in the Appendix A). Isolated yield: 15 % (0.04 g). HRMS calculated for the molecular ion C_21_H_20_: 272.1565; measured: 272.1564. The Raman spectrum is presented in Figure 6 and the main vibrations are reported in Table 2. Unfortunately, this spectrum presents many interferences due to the weak product amount.

2,5-Di(benzyl)toluene (oil)

^1^H NMR (300 MHz, CDCl_3_): δ ppm 7.30–6.99 (m, 13H, arom), 3.95 (s, 2H, CH_2_-Ph), 3.93 (s, 2H, CH_2_-Ph), 2.20 (s, 3H, CH_3_); ^13^C NMR (75 MHz, CDCl_3_): 141.5, 140.7, 139.3, 136.8, 131.0, 130.2, 129.1, 128.9, 128.6, 128.5, 126.6, 126.1, 126.0, 41.7, 39.2, 19.8 (RMN spectra are in the Appendix A). Isolated yield: 15 % (0.04 g). HRMS calculated for the molecular ion C_21_H_20_: 272.1565; measured: 272.1564. The Raman spectrum is presented in Figure 6 and the main vibrations are reported in Table 2.

## 4. Conclusions

Due to the synthesis of 2,4-, 2,5-, 3,4-, and 3,5-dibenzyltoluene and the in-depth analysis of the mass spectra of all isomers, it was revealed that the so-called “DBT” contained isomers of dibenzyltoluene and (benzyl)benzyltoluene; all of the molecules composing the commercial Jarytherm heat transfer oil were identified.

Using fast GC-FID, the degree of hydrogenation was found to be accessible in less than 5 min; moreover, the hydrogenation kinetics of each isomer was made possible. These results represent a real step forward in the study of Hx-DBT hydrogenation and dehydrogenation mechanisms.

## Figures and Tables

**Figure 1 molecules-28-03751-f001:**
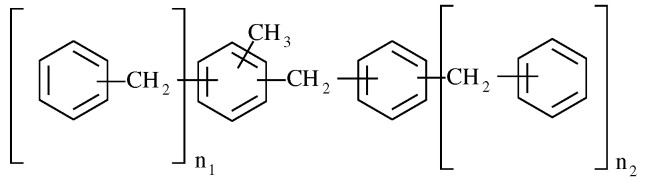
Generic formula of the commercial dibenzyltoluene.

**Scheme 1 molecules-28-03751-sch001:**
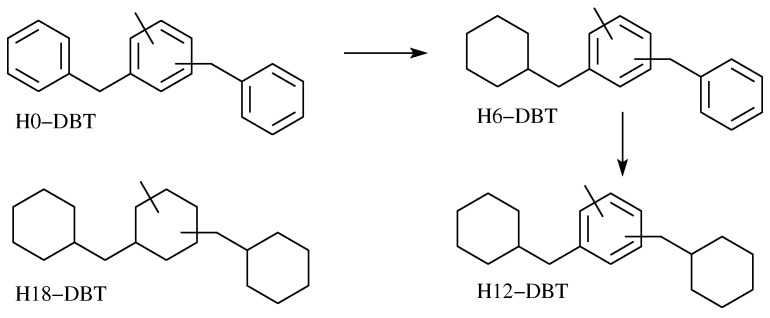
Hydrogenation of H0-DBT using a Ru catalyst.

**Figure 2 molecules-28-03751-f002:**
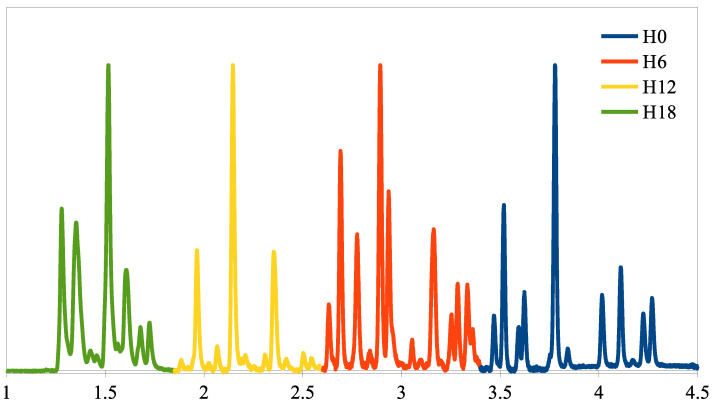
GC-FID chromatogram of a pre-made mixture of H0, H6, H12, and H18-DBT.

**Figure 3 molecules-28-03751-f003:**
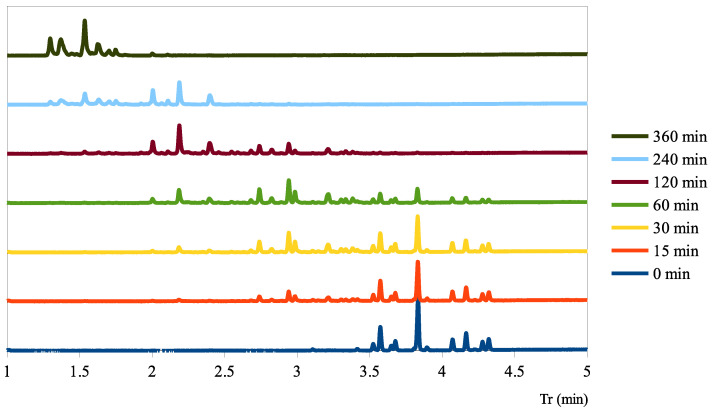
GC-FID chromatograms of samples collected during a hydrogenation follow-up in a batch-stirred tank reactor, at 150 °C, 30 bar H_2_, 1000 rpm using 5 wt-% Pt/C with Pt/H0-DBT = 0.015 mol-%.

**Figure 5 molecules-28-03751-f005:**
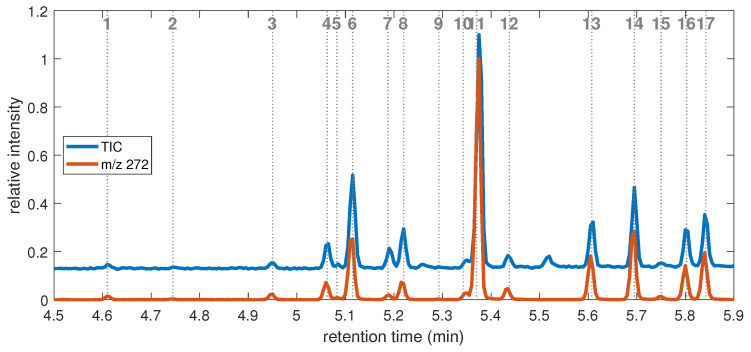
GC–MS chromatogram of the H0-DBT mix zoomed over the 4.5–5.9 min range. Blue (above) represents the total ion count of the chromatogram (TIC) and red (below) represents the extracted ion count for the *m*/*z* 272 of the molecular ion. The 17 observed peaks are identified by their peak numbers.

**Scheme 2 molecules-28-03751-sch002:**
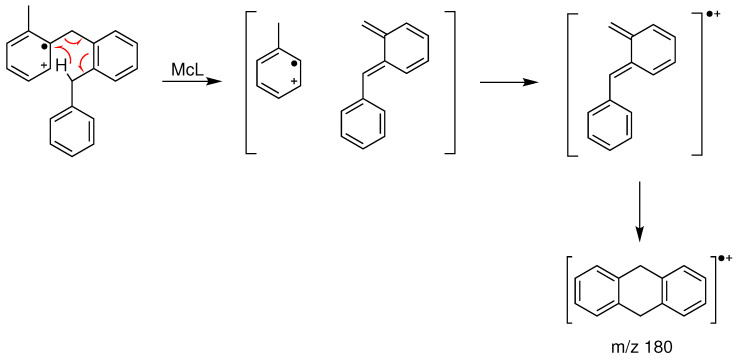
McLafferty rearrangement scheme and formation of the *m*/*z* 180 from the 2,2${}^{\prime }$-BBT.

**Scheme 3 molecules-28-03751-sch003:**
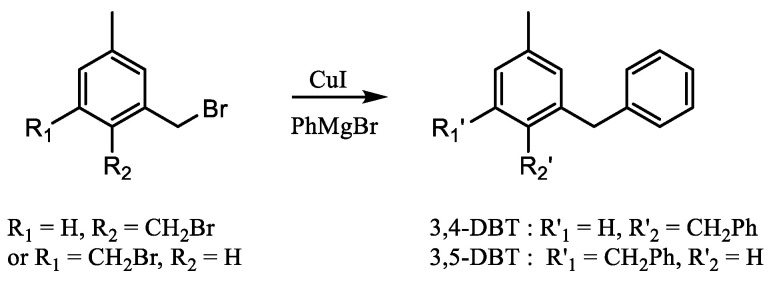
Synthesis of 3,4 and 3,5-dibenzyltoluene.

**Figure 6 molecules-28-03751-f006:**
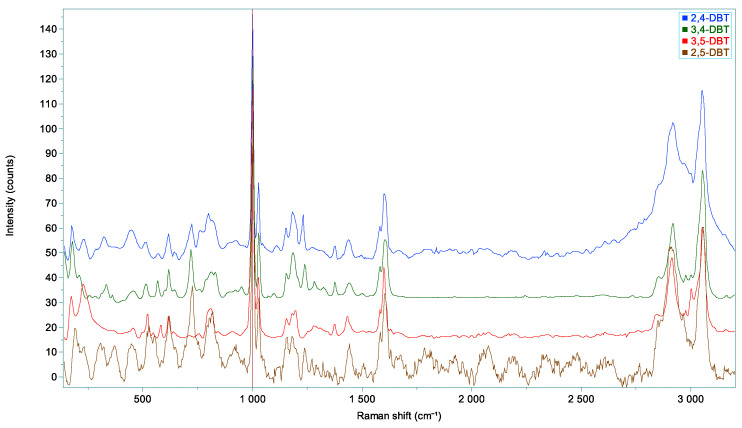
Raman spectra of the four synthesized molecules. Note that the 2,4-DBT spectrum presents many interferences.

**Scheme 4 molecules-28-03751-sch004:**
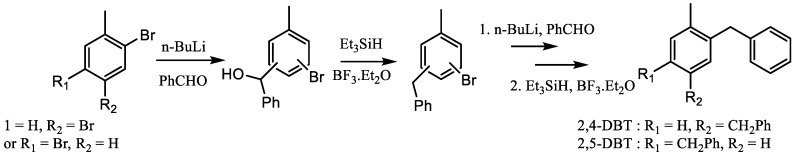
Synthesis of 2,4- and 2,5-dibenzyltoluene.

**Table 2 molecules-28-03751-t002:** Raman vibrations of synthesized molecules compared to those of toluene; vs. very strong, s: strong, m: medium, w: weak, sh: shoulder.

Vibrations	Toluene	2,4-DBT	3,4-DBT	3,5-DBT	2,5-DBT
		173 (m)	176 (m)	173 (m)	
	216 (m)	229 (m)	209 (w)	225 (m)	
		320 (m)	333 (w)		
Aromatic ring deformation vibrations		444 (m)	450 (w)	455 (w)	
	517 (m)	511 (m)	513 (m)	518 (m)	
		568 (w)	567 (m)	577 (m)	
Aromatic out-of-plane C-H deformation vibrations	623 (w)	617 (m)	616 (m)	617 (m)	
		**722 (s)**	**718 (s)**	**-**	**724**
	783 (s)	760 (sh), 798 (m), 816 (sh)	785 (sh), 811 (m), 828 (sh)	809 (m)	818
Aromatic in-plane C-H deformation vibrations	1002 (vs), 1027 (s)	1000 (vs), 1027 (s)	999 (vs), 1026 (s)	1000 (vs), 1027 (s)	1002 (vs), 1027 (s)
Alkane C-C vibrations: skeletal vibrations	1156 (w), 1179 (w)	1150 (m), 1180 (m)	1152 (m), 1185 (m)	1153 (m), 1181 (m)	1179
	1209 (m)	1229 (m)	1238 (m)		1234
Alkane C-H deform. vibrations	1378 (m)	1375 (w)	1375 (w)	1375 (w)	
		1437 (m)	1439 (w)	1431 (m)	1442
Aromatic C=C stretching vibrations	1586 (sh), 1601 (s)	1579 (sh), 1599 (s)	1579 (sh), 1604 (s)	1579 (sh), 1599 (s)	1604 (s)
Alkane C-H stretching Vibrations	2870 (w), 2918 (s), 2982 (sh)	2848 (sh), 2916 (s), 2975 (sh)	2853 (sh), 2918 (s), 2976 (sh)	2843 (sh), 2913 (s), 2975 (sh)	2912 (s)
Aromatic =C-H stretching vibrations	3004, 3056 (s)	3000 (w), 3051 (s)	3000 (w), 3053 (s)	3001 (w), 3051 (s)	3058 (s)

## Data Availability

Data are available from corresponding authors upon request.

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
