# Peer review of "Analysis of Dibenzyltoluene Mixtures: From Fast Analysis to In-Depth Characterization of the Compounds"

_molecules, 2023, doi:10.3390/molecules28093751_

Round 1

Reviewer 1 Report

Although the article is more suitable for the journal of Analytical Chemistry, I’ll recommend it for publication in the journal Molecules. However, there are some remarks. For all synthesized compounds, the yield should be indicated not only as a percentage, but also in grams (as was done for 3,5-DBT). In addition, Raman spectra for the identification of compounds are as important a characteristic as NMR spectra. Therefore, the Raman spectra should be given in the experimental part, and not refer readers to the Supplementary materials.

Author Response

"Although the article is more suitable for the journal of Analytical Chemistry, I’ll recommend it for publication in the journal Molecules. However, there are some remarks. For all synthesized compounds, the yield should be indicated not only as a percentage, but also in grams (as was done for 3,5-DBT)."

We thank the reviewer for this remark. The weights were added.

"In addition, Raman spectra for the identification of compounds are as important a characteristic as NMR spectra. Therefore, the Raman spectra should be given in the experimental part, and not refer readers to the Supplementary materials."

Although it is not a standard request, the Raman spectra, as well as a table with all the characteristic vibrations, were added.

Reviewer 2 Report

In this manuscript, the authors presented a fast-GC analysis strategy to identify the DBT isomers and the corresponding hydrogenation products with different hydrogenation degrees. This work is interesting to the scientific community when studying the LOHC techniques. I recommend that the manuscript can be accepted after the following issues addressed.

(1)  Why the H6, H12 and H18 products should be synthesized at different temperatures?

(2)  In there any overlap of the signals between H6, H12 and H18 products from the GC analysis?  

(3)  Some related publications about the LOHC technology are recommended to referred in the introduction part: Applied Catalysis B: Environmental, 2021, 288, 119996; Journal of Catalysis, 2018, 360, 175-186; International Journal of Hydrogen Energy, 2019, 44, 5345-5354.

Author Response

We would like to thank the reviewer for his/her remarks.

(1)  Why the H6, H12 and H18 products should be synthesized at different temperatures?

The hydrogenation of H0-DBT towards H18-DBT is composed of 3 reactions: H0-DBT to H6-DBT, H6-DBT to H12-DBT and H12-DBT to H18-DBT. Even though these 3 reactions appear as consecutive reactions, they do take place simultaneously in the reaction mixture with different reaction rates. In higher temperatures, the first two hydrogenation reactions are much faster than the last hydrogenation and the molecules are rapidly transformed into a mixture that is rich in H6 or H12-DBT because the rate determining step is the transformation of H12-DBT to H18-DBT. (Catalyst deactivation at higher temperature also plays a role in limiting the final hydrogenation reaction.) Under lower temperatures, even though the global reaction is slower, the formation of H18-DBT can be guaranteed with no side product thanks to milder reaction conditions.

The following sentence was added in the article : "Note that the synthesis of H18-DBT requires a long reaction time (5 h) that is not compatible with the use of high temperatures which would have generated by-products"

(2)  In there any overlap of the signals between H6, H12 and H18 products from the GC analysis?  

There is no overlap of signals between H6, H12 and H18 but there is one overlap of a minor H0 isomer (not DBT nor BBT) species, a PdTm as indicated in SI, which doesn't influence the integration results due to its very small concentration.

(3)  Some related publications about the LOHC technology are recommended to referred in the introduction part: Applied Catalysis B: Environmental, 2021, 288, 119996; Journal of Catalysis, 2018, 360, 175-186; International Journal of Hydrogen Energy, 2019, 44, 5345-5354.

The two first references concern catalysts that were not used for the dehydrogenation of perhydro-dibenzyltoluene, but decalin. It is considered too far from the message carried out by our article. Concerning the last reference, it was added.